# Inlier-Centric Post-Training Quantization for Object Detection Models

**Minsu Kim**[1], **Dongyeun Lee**[1], **Jaemyung Yu**[2], **Jiwan Hur**[1], **Giseop Kim**[3], **& Junmo Kim**[1]
[1]KAIST,     [2]NAVER AI Lab,     [3]DGIST
{minsu.kim, ledoye, jiwan.hur, junmo.kim}@kaist.ac.kr[1],
jaemyung.yu@navercorp.com[2], gsk@dgist.ac.kr[3]

## Abstract

Object detection is pivotal in computer vision, yet its immense computational demands make deployment slow and power-hungry, motivating quantization. However, task-irrelevant morphologies such as background clutter and sensor noise induce redundant activations (or *anomalies*). These anomalies expand activation ranges and skew activation distributions toward task-irrelevant responses, complicating bit allocation and weakening the preservation of informative features. Without a clear criterion to distinguish anomalies, suppressing them can inadvertently discard useful information. To address this, we present InlierQ, an inlier-centric post-training quantization approach that separates anomalies from informative *inliers*. InlierQ computes gradient-aware volume saliency scores, classifies each volume as an inlier or anomaly, and fits a posterior distribution over these scores using the Expectation–Maximization (EM) algorithm. This design suppresses anomalies while preserving informative features. InlierQ is label-free, drop-in, and requires only 64 calibration samples. Experiments on the COCO and nuScenes benchmarks show consistent reductions in quantization error for camera-based (2D and 3D) and LiDAR-based (3D) object detection.

## 1 Introduction

Object detection is a cornerstone of computer vision, enabling applications from autonomous driving to large-scale video analytics. To meet the efficiency demands of on-device deployment, quantization (Chen et al., 2020; Huang et al., 2024) is widely used to reduce computation and power consumption. Yet, quantizing object detectors remains challenging because their activations often mix task-relevant cues with task-irrelevant responses. In particular, detection pipelines evaluate a large number of regions or volumes, many of which do not correspond to true objects, such as cluttered background areas or noisy sensor returns (Zhou et al., 2024). These non-informative responses, which we term *anomalies*, induce diverse activations broadening the activation range and skew it toward task-irrelevant responses (Fig. 1). As conventional quantization (Nagel et al., 2021) minimizes quantization error over all activations, naively applying it to detection models can be adversely affected by anomalies, leading to suboptimal preservation of task-relevant *inliers*.

Although such disruptive anomalies should be filtered out, the lack of a principled way to identify them makes the problem difficult to address. This issue is magnified under low-bit quantization, where only a few bit levels are available and each level covers a wide value range. When anomalies broaden the layer-wise activation range, the coarse levels in the low-bit quantization regime have to cover task-irrelevant anomalies and relevant *inliers* simultaneously. As a result, anomalies can disproportionately influence the quantization objective, leaving insufficient resolution to faithfully represent task-relevant *inliers*. This challenge remains insufficiently addressed in quantization for object detection models (Chen et al., 2020; Huang et al., 2024; Zhou et al., 2024).

To resolve these problems, we propose Inlier-Centric post-training Quantization (InlierQ), which separates inliers and focuses quantization error reduction on them. For a given volume, which denotes a pixel in 2D and a voxel in 3D, InlierQ assigns an inlier probability, and interprets high-probability volumes as inliers and low-probability ones as anomalies. Concretely, InlierQ computes gradient-aware volume saliency scores from object heatmap (Duan et al., 2019) activations, since

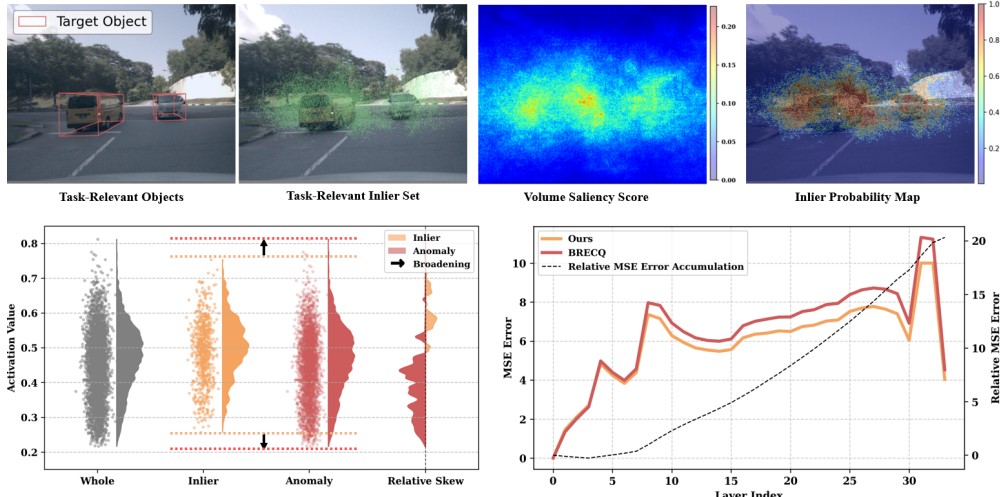

Figure 1: **Effects of Inlier-Centric Quantization.** (Top) InlierQ computes a gradient-aware volume saliency score, estimates an inlier probability map, and samples high-probability volumes to form a task-relevant inlier set (green points) around target objects. (Bottom) Kernel density estimates with scattered activations and the relative skew show that anomalies broaden and skew the activation distribution. By restricting quantization to inliers, InlierQ yields a compressed activation range while preserving task-relevant information. This leads to lower layer-wise MSE, slower relative error accumulation, and up to 3.2% mAP improvement over a baseline method (Li et al., 2021).

these heatmaps explicitly encode object location and class confidence, allowing the criterion to reflect task relevance rather than raw activation magnitude. To make the inlier and anomaly split explicit, InlierQ models the saliency scores as a two-component probabilistic mixture estimated by the Expectation-Maximization (EM) algorithm (Dempster et al., 1977). Using the volume-wise probabilities, we define the inlier set, and InlierQ then selects a quantization range that minimizes quantization error on inliers. Our InlierQ regime alleviates anomaly-induced distributional skew and range inflation, reducing layer-wise activation quantization errors while preserving task-relevant information. We evaluate InlierQ on the COCO and nuScenes object detection benchmarks and show scalable gains across sensors (camera and LiDAR) and modalities (2D and 3D), achieving up to +0.4% mAP on 2D and +3.2% mAP on 3D under W4A4 (INT4) post-training quantization.

## 2 RELATED WORK

**Quantization.** Model quantization is generally divided into Quantization-Aware Training (QAT) (Gong et al., 2019; Choi et al., 2018; Esser et al., 2019; Li et al., 2019) and Post-Training Quantization (PTQ) (Nagel et al., 2019; 2020a; Li et al., 2021). QAT simulates quantization effects during training but requires heavy computation. While QAT explicitly simulates quantization effects, it adds substantial compute. PTQ, by contrast, frequently suffices for 8-bit; data-free approaches have likewise been studied (Nagel et al., 2019). Accurate low-bit PTQ has also become feasible, as shown by Adaround (Nagel et al., 2020a) and further improved by Li et al. (Li et al., 2021).

**Object Detection Models.** Several works have proposed post-training quantization for object detection models. AQD (Chen et al., 2020) introduces a fully quantized detector, while HQOD (Huang et al., 2024) proposes a harmonious strategy to balance accuracy and efficiency. LiDAR-PTQ (Zhou et al., 2024) introduces post-training quantization for sparse 3D LiDAR object detection models. Although these methods mitigate the influence of anomalies, they lack a principled way to isolate them and thus still struggle with anomaly-driven redundant activations.

**Outlier Suppression.** A related line of research targets *outliers*, rare activation values with unusually large magnitude, and suppresses their impact to stabilize quantization. SmoothQuant (Yao et al., 2022) softens outliers via per-channel scaling, while SVDQuant (Li et al., 2024) suppresses high-energy activation components. DMQ (Lee et al., 2025) further introduces learned per-channel scaling. Overall, outlier suppression reduces the dominance of extreme magnitudes, whereas our approach identifies task-irrelevant anomalies and excludes them during quantization.

---

**Algorithm 1** Inlier-Centric Quantization

---

**Input:** Pretrained FP model; Calibration dataset $\mathbf{V}$; Bit-width $b$; Threshold $\tau$; Iteration $T$
**Output:** Quantization parameter set across all layers $S = \{S^1, \ldots, S^L\}$

1: **for** each layer $l = 1, \ldots, L$ in the pretrained FP model **do**
2:      **for** each calibration sample in $\mathbf{V}$ **do**
3:          Collect full-precision input and output activations;
4:          Identify the inlier set $\mathcal{I}$ (cf. eq. (11));
5:          Update $S^l$ using min-max calibration;
6:      **end for**
7:      **for** optimization step $t = 1, \ldots, T$ **do**
8:          Refine $S^l$ by minimizing the inlier-aware objective (eq. (8));
9:      **end for**
10: **end for**
11: **Return:** Quantized model with parameter set $S = \{S^1, \ldots, S^L\}$;

---

## 3   PRELIMINARIES

Given $b$ bits, quantization categorizes any number of full-precision data $x \in \mathbb{R}$ into a predefined set $\mathcal{B} = \{x_q \mid x_q \in \{x_0, \ldots, x_r\}\}$, where $r = 2^b - 1$ denotes the maximum representable index under $b$-bit precision. The lower and upper bounds of the integer code range are denoted by $l$ and $u$, respectively, with $u - l + 1 = 2^b$. To this end, the quantization operation deploys the following:

$$x_q = \text{Clamp}\left(\left\lfloor \frac{x}{s} \right\rceil + z;\; l,\; u\right),\tag{1}$$

$$\text{where} \quad \text{Clamp}(x;\; l, u) = \begin{cases} l, & x < l, \\ x, & l \le x \le u, \\ u, & x > u, \end{cases}$$

$\lfloor \cdot \rceil$ denotes rounding to the nearest element of $\mathcal{B}$. Following this convention of quantization (Nagel et al., 2021), the parameter set $S = \{s, z\}$ (scale and zero-point) is employed. The extent to which representational capacity is preserved strongly depends on the fidelity of $S$. Post-training quantization (PTQ) aims to find layer-wise optimal $S$ for activations or weights.

**Optimization-based PTQ.** The objective of quantization can be formalized as minimizing the expected change in task loss $\mathcal{L}$ induced by the activation perturbation $\Delta \mathbf{x}_S = \mathbf{x}_q - \mathbf{x}$ between a full precision (FP32) $\mathbf{x}$ and its quantized counterpart $\mathbf{x}_q$ as follows:

$$\arg\min_S \; \mathbb{E}\big[\mathcal{L}(\mathbf{x} + \Delta\mathbf{x}_S) - \mathcal{L}(\mathbf{x})\big].\tag{2}$$

Based on eq. (2), Nagel et al. (2020b) introduced a Taylor expansion of the loss with respect to the activations for approximating the quantization objective:

$$\arg\min_S \; \mathbb{E}[\mathcal{L}(\mathbf{x} + \Delta\mathbf{x}_S) - \mathcal{L}(\mathbf{x})] \approx \arg\min_S \; \Delta\mathbf{x}_S \cdot \mathbf{g}^{(\mathbf{x})\top} + 0.5 \cdot \mathbb{E}\Big[\Delta\mathbf{x}_S^\top \cdot \mathbf{H}^{(\mathbf{x})} \cdot \Delta\mathbf{x}_S\Big],\tag{3}$$

where $\mathbf{H}^{(\mathbf{x})} \triangleq \nabla_\mathbf{x}^2 \mathcal{L}(\mathbf{x})$ denotes the Hessian of the objective with respect to the activation vector $\mathbf{x}$, and $\mathbf{g}^{(\mathbf{x})} \triangleq \nabla_\mathbf{x}\mathcal{L}(\mathbf{x})$ denotes the corresponding activation gradient. Building on this, Li et al. (2021) adopts the common approximation that for (negative) log-likelihood objectives, the expected Hessian coincides with the Fisher Information Matrix (FIM):

$$\mathbb{E}\big[\mathbf{H}^{(\mathbf{x})}\big] = \mathbb{E}\big[\mathbf{g}^{(\mathbf{x})}\mathbf{g}^{(\mathbf{x})\top}\big].\tag{4}$$

Overall, for a loss function $\mathcal{L}$ derived from a probabilistic model, the PTQ optimization is governed by the activation perturbation induced by quantization, $\Delta\mathbf{x}_S$, together with the activation gradients $\mathbf{g}^{(\mathbf{x})}$. In particular, under (negative) log-likelihood objectives, the expected Hessian coincides with the FIM, which can be expressed via gradient outer products in expectation.

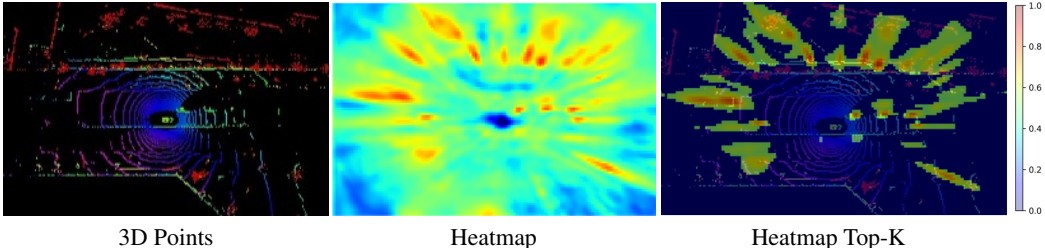

|  |  |  |
|:---:|:---:|:---:|
| 3D Points | Heatmap | Heatmap Top-K |

Figure 2: **3D points (left), predicted heatmap (middle), and heatmap top-$K$ overlaid on 3D points (right).** In constructing the loss function eq. (7), the top-$K$ heatmap predictions are exploited to prioritize preserving model performance in object searching.

## 4 METHOD

### 4.1 PROBLEM FORMULATION

When applying optimization-based PTQ to object detection models, the scheme faces inherent limitations because it optimizes all activations uniformly, without regard to their task relevance. As shown in Fig. 1, the non-uniform distributions amplify quantization error, particularly due to noisy or high-valued anomalies. In this section, we suggest a novel method to address the problem.

**Decomposed Space.** We first partition the volume space $\mathcal{V}$ into an inlier region $\mathcal{I}$ and an anomaly region $\mathcal{A} := \mathcal{V} \setminus \mathcal{I}$, such that $\mathcal{V} = \mathcal{I} \cup \mathcal{A}$. This partition captures the fact that not all activations contribute equally to the optimization objective: inliers typically encode task-relevant information, whereas anomalies often introduce noise or distort quantization. Crucially, this separation allows us to assign different responsibilities to $\mathcal{I}$ and $\mathcal{A}$, so that the effect of anomalous activations can be explicitly isolated rather than implicitly mixed into the overall objective. Based on this decomposition, the optimization problem can be reformulated as:

$$\arg\min_S \; \mathbb{E}[f(\mathbf{x}, \mathbf{x}_q; \mathbf{w})] = \sum_{\mathcal{D} \in \{\mathcal{I}, \mathcal{A}\}} \lambda_{\mathcal{D}} \cdot \mathbb{E}_{\mathbf{x} \sim \mathcal{D}}[f(\mathbf{x}, \mathbf{x}_q; \mathbf{w})], \tag{5}$$

$$\text{where} \quad f(\mathbf{x}, \mathbf{x}_q; \mathbf{w}) = \mathcal{L}(\mathbf{x} + \Delta\mathbf{x}_S; \mathbf{w}) - \mathcal{L}(\mathbf{x}; \mathbf{w}),$$

$f$ denotes a function that measures the loss difference induced by an activation perturbation, and $\lambda_{\mathcal{D}}$ denotes the responsibility for region $\mathcal{D} \in \{\mathcal{I}, \mathcal{A}\}$. The objective in eq. (5) measures the expected change in the loss $\mathcal{L}$ caused by the quantization induced activation error $\Delta\mathbf{x}_S$. To make the problem tractable, we approximate $f(\mathbf{x}, \mathbf{x}_q; \mathbf{w})$ using a Taylor series expansion around $\mathbf{x}$, as introduced in section 3. Following prior works (Nagel et al., 2020a; Li et al., 2021), we assume that $\Delta\mathbf{x}$ is approximately unbiased and weakly correlated with $\mathbf{g}$, so that $\mathbb{E}[\mathbf{g}^\top \Delta\mathbf{x}_S] \approx 0$. Under this condition, the objective is governed by the Hessian $\mathbf{H}^{(\mathbf{x})}$ of $\mathcal{L}$ with respect to $\mathbf{x}$:

$$\arg\min_S \; \sum_{\mathcal{D} \in \{\mathcal{I}, \mathcal{A}\}} \lambda_{\mathcal{D}} \cdot \mathbb{E}_{\mathbf{x} \sim \mathcal{D}}\big[\Delta\mathbf{x}_S^\top \cdot \mathbf{H}^{(\mathbf{x})} \cdot \Delta\mathbf{x}_S\big], \tag{6}$$

**Loss function.** To maximize the preservation of object-related information, we design a loss that emphasizes salient activations by restricting curvature computation to heatmap top-$K$ entries. We minimize the negative log-likelihood over the top-$K$ heatmap values across all channels:

$$\mathcal{L}(\mathbf{x}; \mathbf{w}) = -\frac{1}{KC} \sum_{k=1}^{K} \sum_{c=1}^{C} \log \mathcal{H}_{[k],c}, \tag{7}$$

where $\mathcal{H}_{[k],c}$ denotes the $k$-th largest heatmap value in a channel (or class index) $c$ (See Fig. 2). This loss serves as an auxiliary objective, enabling the computation of gradients and Hessians with respect to the quantization perturbation $\Delta\mathbf{x}$ while focusing only on the most informative activations. In the appendix, we prove that, under eq. (7), the expected Hessian equals to the FIM, and in section 5, we empirically investigate whether the anomaly space $\mathcal{A}$ impedes quantization error minimization while demonstrating that the inlier space $\mathcal{I}$ serves as a sufficient subspace for generalization:

$$\mathbb{E}_{\mathbf{x} \sim \mathcal{V}}[f(\mathbf{x}, \mathbf{x}_q; \mathbf{w})] \approx \mathbb{E}_{\mathbf{x} \sim \mathcal{I}}\big[f(\mathbf{x}, \mathbf{x}_q; \mathbf{w})\big]. \tag{8}$$

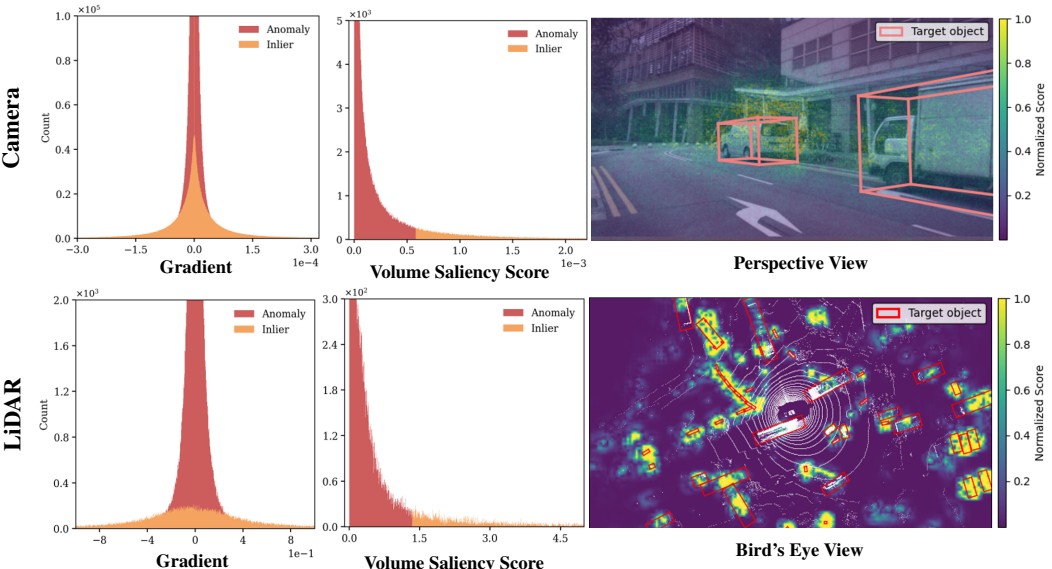

Figure 3: **Cross-modal Behavior of Gradients and Volume Saliency Scores.** (Left) Gradient-domain histograms indicate that each modality has different inlier distributions. (Middle) Yet, in the space of volume saliency, both modalities exhibit a consistent, modality-invariant distribution. (Right) Volume saliency scores overlaid on the labeled input signals highlight objects while appropriately modulating the contribution of background, serving as a task-relevant scoring function.

## 4.2 VOLUME SALIENCY MODEL

**Volume Saliency Score.** We define the volume saliency score $h(\mathbf{x})$ as the L1 norm of the loss gradient with respect to the activation vector:

$$h(\mathbf{x}) = \sum_{m=1}^{C} \left| \frac{\partial \mathcal{L}(\mathbf{x}; \mathbf{w})}{\partial \mathbf{x}_m} \right|. \tag{9}$$

This score measures how sensitive the loss is to perturbations along different activation dimensions and highlights the channels that most strongly influence the task objective. By capturing the aggregated top-$K$ responses, $h(\mathbf{x})$ emphasizes dominant volumetric regions while naturally suppressing less informative ones, thereby inducing the partition of the gradient domain illustrated in Fig. 3.

**Posterior.** We further formulate a probabilistic model that interprets the volume saliency score in terms of inlier likelihood. Specifically, by referencing this, we construct a posterior distribution indicating whether the score of $\mathbf{x}$ belongs to the inlier space $\mathcal{I}$ or the anomaly space $\mathcal{A}$. The posterior is modeled as:

$$P(\mathcal{I} \mid h(\mathbf{x})) = \frac{P(h(\mathbf{x}) \mid \mathcal{I}) \cdot P(\mathcal{I})}{\sum_{\mathcal{D} \in \{\mathcal{I}, \mathcal{A}\}} P(h(\mathbf{x}) \mid \mathcal{D}) \cdot P(\mathcal{D})}. \tag{10}$$

Here, $P(h(\mathbf{x})|\mathcal{D})$ denotes the likelihood of observing $h(\mathbf{x})$ under partition $\mathcal{D}$, while $P(\mathcal{D})$ specifies the prior probability of each partition $\mathcal{D} \in \{\mathcal{I}, \mathcal{A}\}$. In practice, we fit a two-component Gaussian mixture model using the Expectation–Maximization (EM) algorithm, where one component represents the inlier distribution and the anomaly distribution. This formulation enables layer-wise scoring of feature vectors, allowing them to be classified as either inliers or anomalies.

**Inlier Set Definition.** Based on the posterior probability in eq. (10), we define the inlier set $\mathcal{I}$ as the collection of feature vectors whose saliency scores are sufficiently likely to belong to the inlier distribution. Formally,

$$\mathcal{I} := \{\mathbf{x} \mid P(\mathcal{I} \mid h(\mathbf{x})) \geq \tau\}, \tag{11}$$

where $\tau$ is a predefined threshold that controls the trade-off between including uncertain activations and filtering out potential anomalies. This thresholding rule ensures that only activations with high inlier posterior probability are retained for quantization, thereby refining the effective feature

Table 1: **Quantitative comparisons with baselines.** We evaluate on the 2D and 3D object detection benchmarks under various bit precision settings. $Wn_1An_2$ denotes $n_1$-bit weight and $n_2$-bit activation quantization. "(C)" and "(L)" indicates camera and LiDAR modality backbones, respectively.

(a) **2D Detection.**

| Detector | Backbone | Bits | Method | $AP_{50}$ | $AP_{75}$ | $mAP_s$ | $mAP_m$ | $mAP_l$ | mAP |
|---|---|---|---|---|---|---|---|---|---|
| RetinaNet | RegNet-3.2GF (C) | FP32 | - | 58.4 | 41.9 | 22.6 | 43.5 | 50.8 | 39.0 |
| | | W8A8 | BRECQ | 58.2 | 41.7 | 22.4 | 43.3 | 50.8 | 38.9 |
| | | W8A8 | LiDAR-PTQ | **58.3** | 41.7 | **22.5** | 43.3 | **50.9** | **39.0** |
| | | W8A8 | Ours | **58.3** | **41.8** | 22.5 | **43.4** | 50.8 | **39.0** |
| | | W4A8 | BRECQ | **55.0** | 38.6 | 20.2 | **39.6** | **47.5** | **36.4** |
| | | W4A8 | LiDAR-PTQ | 54.8 | 38.7 | **20.3** | 39.5 | 47.3 | 36.3 |
| | | W4A8 | Ours | **55.0** | **38.8** | 20.2 | 39.5 | **47.5** | **36.4** |
| | | W4A4 | BRECQ | 52.1 | 36.1 | 18.5 | 36.7 | 44.9 | 34.0 |
| | | W4A4 | LiDAR-PTQ | **52.6** | 36.5 | **18.9** | **37.8** | 44.8 | 34.4 |
| | | W4A4 | Ours | **52.6** | **36.7** | 18.6 | 37.5 | **46.1** | **34.7** |
| Faster R-CNN | ResNet-50 (C) | FP32 | - | 58.1 | 41.2 | 21.6 | 41.6 | 49.1 | 37.9 |
| | | W8A8 | BRECQ | 58.0 | 41.1 | 21.4 | 41.5 | **49.3** | **37.8** |
| | | W8A8 | LiDAR-PTQ | 57.9 | **41.2** | 21.2 | **41.6** | 49.2 | 37.7 |
| | | W8A8 | Ours | **58.1** | 41.1 | **21.5** | **41.6** | 48.9 | **37.8** |
| | | W4A8 | BRECQ | 55.8 | 38.9 | 19.6 | **39.6** | 47.2 | 36.0 |
| | | W4A8 | LiDAR-PTQ | 55.7 | 38.9 | 19.6 | 39.5 | 47.2 | 36.0 |
| | | W4A8 | Ours | **55.9** | **39.2** | **19.8** | **39.6** | **47.4** | **36.1** |
| | | W4A4 | BRECQ | 52.1 | 35.1 | 18.0 | 36.6 | 42.6 | 32.7 |
| | | W4A4 | LiDAR-PTQ | 54.0 | 36.7 | 19.0 | 37.6 | 44.6 | 34.3 |
| | | W4A4 | Ours | **54.7** | **37.3** | **19.2** | **38.3** | **45.2** | **34.7** |

(b) **3D Detection.**

| Detector | Backbone | Bits | Method | mATE | mASE | mAOE | mAVE | mAAE | mAP | NDS |
|---|---|---|---|---|---|---|---|---|---|---|
| DETR3D | ResNet-101 (C) | FP32 | - | 0.778 | 0.274 | 0.442 | 0.851 | 0.202 | 33.8 | 41.4 |
| | | W8A8 | BRECQ | 0.778 | **0.274** | **0.438** | 0.852 | 0.203 | 33.5 | 41.2 |
| | | W8A8 | LiDAR-PTQ | **0.777** | 0.276 | 0.441 | **0.848** | 0.203 | 33.5 | **41.3** |
| | | W8A8 | Ours | **0.777** | **0.274** | 0.444 | 0.852 | **0.201** | 33.6 | **41.3** |
| | | W4A8 | BRECQ | 0.814 | 0.282 | 0.499 | 0.949 | 0.218 | 29.0 | 36.9 |
| | | W4A8 | LiDAR-PTQ | 0.813 | 0.284 | 0.494 | 0.956 | 0.222 | 28.8 | 36.8 |
| | | W4A8 | Ours | **0.812** | **0.280** | **0.488** | **0.945** | **0.217** | **29.1** | **37.1** |
| | | W4A4 | BRECQ | 0.866 | 0.285 | 0.523 | 0.962 | 0.232 | 24.8 | 33.8 |
| | | W4A4 | LiDAR-PTQ | 0.850 | 0.287 | 0.539 | 0.967 | 0.232 | 25.2 | 34.0 |
| | | W4A4 | Ours | **0.840** | **0.283** | **0.495** | **0.946** | **0.230** | **26.4** | **35.2** |
| CenterPoint | VoxelNet (L) | FP32 | - | 0.288 | 0.254 | 0.326 | 0.282 | 0.187 | 56.3 | 64.8 |
| | | W8A8 | BRECQ | 0.298 | **0.257** | 0.348 | 0.294 | **0.182** | 54.4 | 63.4 |
| | | W8A8 | LiDAR-PTQ | 0.299 | 0.258 | 0.353 | **0.290** | 0.183 | 54.4 | 63.4 |
| | | W8A8 | Ours | **0.297** | **0.257** | **0.346** | 0.293 | **0.182** | **54.7** | **63.6** |
| | | W4A8 | BRECQ | 0.309 | 0.261 | 0.378 | 0.309 | 0.193 | 51.7 | 61.4 |
| | | W4A8 | LiDAR-PTQ | 0.309 | 0.260 | 0.390 | 0.309 | **0.187** | 50.7 | 60.8 |
| | | W4A8 | Ours | **0.304** | **0.259** | **0.376** | **0.304** | 0.192 | **52.2** | **61.7** |
| | | W4A4 | BRECQ | 0.324 | 0.263 | 0.406 | 0.356 | 0.193 | 43.4 | 56.3 |
| | | W4A4 | LiDAR-PTQ | 0.326 | 0.272 | 0.431 | 0.366 | 0.192 | 39.5 | 54.0 |
| | | W4A4 | Ours | **0.319** | **0.261** | **0.401** | **0.343** | **0.189** | **46.6** | **58.1** |

space and reducing the adverse influence of noisy or inlier volumes. In practice, $\tau$ can be selected empirically or calibrated using validation data to balance precision and recall of the inlier set.

**Inlier-Centric Quantization.** With our extraction of the inlier set $\mathcal{I}$, we optimize below:

$$\arg \min_S \; \mathbb{E}_{\mathbf{x} \in \mathcal{I}} \left[ \Delta \mathbf{x}_S^\top \cdot \mathbf{H}^{(\mathbf{x})} \cdot \Delta \mathbf{x}_S \right]. \tag{12}$$

This formulation explicitly discards anomalous activations and focuses only on the curvature of inlier distributions, thereby suppressing the destabilizing effect of anomalies and guiding the quantization process toward semantically meaningful volumes.

## 5 EXPERIMENTS

**Datasets.** We evaluate the proposed method on both 2D and 3D object detection benchmarks. For 2D detection, we use the COCO dataset (Lin et al., 2014), which provides 80 object categories and challenging real-world scenes. For 3D detection, we adopt the nuScenes dataset (Caesar et al.,

2020), a large-scale object detection benchmark with 10 object categories. During the calibration of both 2D and 3D object detection models, we use 64 samples from the training split. For evaluation, all reported results are obtained on the full validation sets of each benchmark, ensuring a fair comparison with prior work. To isolate the effect of quantization, we exclude all data augmentation.

**Baseline PTQs.** We adopt BRECQ (Li et al., 2021) and LiDAR-PTQ (Zhou et al., 2024), applied to the same architectures and calibration configurations for fair comparison. BRECQ minimizes Hessian-informed reconstruction error through block-wise feature alignment. LiDAR-PTQ, originally designed for LiDAR-based 3D detection, employs both bounding box regression loss and classification loss to align quantized outputs with task objectives. Although developed for LiDAR, its optimization method is universal, and we therefore also adopt it as a baseline for the quantization of camera-based models. All the methods follow a layer-wise sequential optimization scheme.

**Implementation Details.** We evaluate our method on four architectures: Faster R-CNN (Ren et al., 2015) with ResNet-50 (He et al., 2016) backbone, and RetinaNet (Lin et al., 2017) with RegNet-3.2GF (Radosavovic et al., 2020) backbone for 2D object detection, DETR3D (Wang et al., 2022) for camera-based 3D detection, and CenterPoint (Yin et al., 2021) for LiDAR-based 3D detection. These serve as the target models for quantization in our study. For quantization, we first apply weight quantization using min–max scaling, followed by activation quantization. The first layer of Faster R-CNN and DETR3D is quantized to INT8, while the first layer of CenterPoint is quantized to FP16. Moreover, for all models, the detection head is retained in FP16 to prevent severe degradation of task performance and to isolate the quantization effects under investigation (Li et al., 2021).

## 5.1 2D OBJECT DETECTION

**Evaluation Metric.** We follow the standard COCO detection metrics for 2D object detection. Specifically, we report $AP_{50}$ and $AP_{75}$, which correspond to the average precision at IoU thresholds of 0.50 and 0.75, respectively, thereby reflecting detection accuracy under loose and strict localization criteria. In addition, we evaluate scale-specific performance using $mAP_s$, $mAP_m$, and $mAP_l$, which measure the mean AP for small, medium, and large objects, respectively, to capture scale sensitivity. Finally, we provide the overall mAP, computed as the mean AP averaged over IoU thresholds from 0.50 to 0.95 with a step size of 0.05, which serves as the primary metric summarizing overall detection performance.

**Comparisons.** Table. 1a summarizes the results under different quantization settings, where we compare against BRECQ (Li et al., 2021) and LiDAR-PTQ (Zhou et al., 2024). In the W8A8 setting, RetinaNet and Faster R-CNN show comparable performance to the full-precision model, with 37.8%, 37.7%, and 37.8% mAP for BRECQ, LiDAR-PTQ, and ours, indicating that 8-bit precision already suffices to cover anomaly distributions. In the more aggressive W4A8 setting, our method achieves 36.1% and 36.4% mAP on RetinaNet and Faster R-CNN, slightly outperforming BRECQ (36.0%, 36.4%) and LiDAR-PTQ (36.0%, 36.3%), showing a small but consistent gain despite the expected degradation from lower weight precision. The gap becomes more pronounced in the W4A4 regime, where our method reaches 34.7% mAP for both backbones, clearly surpassing BRECQ (32.7%, 34.0%) and LiDAR-PTQ (34.3%, 34.4%). Overall, these results show that quantizing activations on the compressed, de-skewed inlier distributions preserves and often improves accuracy compared to the baselines, especially under 4-bit activations. This indicates that the rejected anomalies carry little task-relevant information and that the proposed inlier set is a safe target for quantization. In the 8-bit regime, the gains are modest, whereas in the more aggressive 4-bit setting, they become pronounced because representing the entire activation distribution, including anomalies, is much harder and consumes a disproportionate fraction of the limited quantization levels. This behavior directly supports the inlier-centric design, where removing anomalies is particularly beneficial at low bit precision.

## 5.2 3D OBJECT DETECTION

**Evaluation Metric.** We adopt the official nuScenes evaluation protocol. The performance is measured using the following metrics: mATE (mean Average Translation Error, measuring the average distance between predicted and ground-truth box centers), mASE (mean Average Scale Error, quantifying the discrepancy in object size), mAOE (mean Average Orientation Error, evaluating heading misalignment), mAVE (mean Average Velocity Error, capturing velocity prediction error), and mAAE (mean Average Attribute Error, reflecting the accuracy of attribute estimation such as object

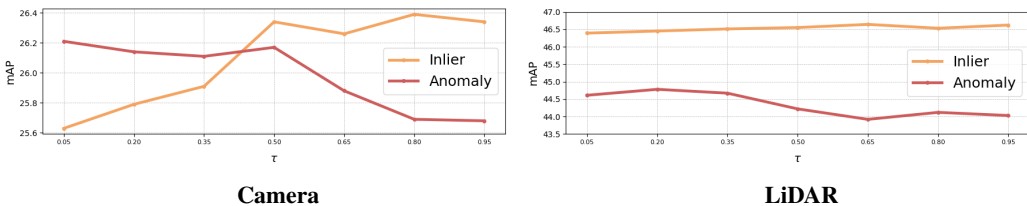

**Figure 4: For varying threshold strictness.** X-axis corresponds to the threshold parameter in eq. (11), and Y-axis is task performance. Inlier (or Anomaly) set defined with a higher $\tau$ applies a stricter (or less strict) decision, causing performance gain (or drop).

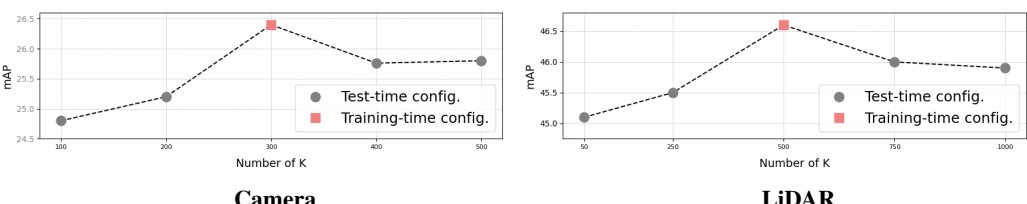

**Figure 5: For varying Heatmap Top-K.** X-axis indicates the number of heatmap Top-K parameter, and Y-axis is the task performance.

state). In addition, we report the 3D mAP, defined over multiple distance-based thresholds, and the nuScenes Detection Score (NDS), which combines mAP with the aforementioned error metrics into a single indicator. In this subsection, we use the notation (x%, y) as x% mAP and y NDS.

**Comparisons on Camera Modality.** Under the W8A8 setting, all methods perform comparably to the full-precision baseline. Our method achieves (33.6%, 41.3), which is on par with BRECQ (33.5%, 41.2) and LiDAR-PTQ (33.5%, 41.3). This overall trend is consistent with the results observed in camera-based 2D object detection. When moving to W4A8 quantization, performance degradation becomes more evident. Our method attains (29.1%, 37.1), slightly higher than BRECQ (29.0%, 36.9) and LiDAR-PTQ (28.8%, 36.8). The performance gap becomes more pronounced in the most challenging W4A4 setting with 4-bit activations, where our approach yields (26.4%, 35.2), clearly surpassing BRECQ (24.8%, 33.8) and LiDAR-PTQ (25.2%, 34.0). These results confirm that inlier-centric quantization improves robustness against aggressive precision reduction in camera-based 3D detection, with especially notable gains under low-bit activation constraints.

**Comparisons on LiDAR Modality.** For LiDAR detection, the performance trend differs from the camera modality, as accuracy degradation is already observed at 8-bit quantization. This can be attributed to the default precision of nuScenes LiDAR points, which store reflectance in 16-bit and distance in 8-bit formats, whereas camera inputs are typically represented in uint8. Consequently, approximating LiDAR features at lower 4 and 8 bit-widths induces larger errors, leading to relatively greater performance loss compared to the camera case. In the W8A8 setting, our method achieves (55.7%, 63.6), closely matching BRECQ (55.4%, 63.4) and LiDAR-PTQ (55.7%, 63.5). At W4A8, our method shows a clearer advantage, reaching (52.2%, 61.7) compared to BRECQ (51.7%, 61.4) and LiDAR-PTQ (52.0%, 61.6). Finally, under the most constrained W4A4 setting, our approach achieves (46.6%, 58.1), significantly outperforming BRECQ (39.5%, 54.2) and LiDAR-PTQ (42.2%, 56.3).

## 5.3 SPACE DECOMPOSITION

In Fig. 4, we gradually increase the threshold parameter $\tau$ on both the inlier and anomaly probability distributions, construct the corresponding inlier and anomaly sets, and then perform quantization using only the activations in each set. As illustrated, a larger $\tau$ enforces a stricter decision when classifying samples as inliers or anomalies, which correspondingly raises or lowers the performance of the inlier and anomaly distributions, respectively, and yields a smooth, largely monotonic transition in performance. These trends are consistently observed for both camera and LiDAR modalities, indicating that the posterior distributions reliably capture inlier and anomaly samples, assign each sample to one of the two distributions in a stable and interpretable manner, and thus provide a meaningful basis for our inlier-centric quantization scheme.

## 5.4 ABLATION STUDY

**Heatmap Top-K.** In Fig. 5, we report 3D object detection performance of InlierQ as a function of the heatmap top-$K$ parameter in eq. (7). We gradually increase $K$ from 100 to 500 for the camera modality and from 50 to 1000 for the LiDAR modality. Note that DETR3D employs top-300 heatmap queries during training, while CenterPoint uses top-500 queries; these training-time configurations are highlighted in red, and the others in gray. As shown in the figure, increasing $K$ generally improves the performance of the quantized model up to the red training-time $K$, beyond which performance starts to decline. Although a larger $K$ provides richer information by incorporating more heatmap samples, an excessively large $K$ introduces many task-irrelevant regions that contaminate the inlier set and weaken the inlier-centric optimization. Overall, this analysis shows that the training-time choice of $K$ offers a good trade-off between information richness and noise, and yields the most effective definition of task-relevant inlier sets for InlierQ.

**Inlier Set.** Table 2 presents an ablation study analyzing the contributions of heatmap top-K selection, inlier distributions, and anomaly distributions across 2D/3D object detection tasks and modalities. Comparing anomaly-only optimization with and without heatmap top-K (the first vs. second rows for each modality) shows consistent gains, yielding improvements of +2.0%, +1.0%, and +1.5% mAP for 2D, 3D camera, and 3D LiDAR detection, respectively, which confirms that focusing anomaly modeling on high-confidence regions is beneficial. For the remaining comparisons, including anomalies consistently degrades performance, with drops of -0.1% for the 2D camera, -0.3% for the 3D camera, and -0.6% mAP for 3D LiDAR detection, indicating that anomalies hinder the effectiveness of the inlier-centric optimization and dilute the benefits of concentrating quantization on task-relevant activations. Despite their adverse effect, the anomaly distribution shares similar data statistics with the inlier set, such as comparable min/max ranges and intensity distributions. This statistical resemblance prevents a severe collapse of task performance, suggesting that anomalies behave as mildly harmful perturbations rather than entirely unstructured noise, and explaining why InlierQ can still retain reasonable accuracy even when anomalies are not fully removed.

Table 2: **Ablation Study on Proposed Methods.**

| Task | Modality | Heatmap | Inlier | Anomaly | mAP |
|------|----------|---------|--------|---------|-----|
| 2D Det. | Camera | - | - | ✓ | 32.5 |
|         |        | ✓ | - | ✓ | 34.5 |
|         |        | ✓ | ✓ | - | **34.7** |
|         |        | ✓ | ✓ | ✓ | 34.6 |
| 3D Det. | Camera | - | - | ✓ | 24.8 |
|         |        | ✓ | - | ✓ | 25.8 |
|         |        | ✓ | ✓ | - | **26.4** |
|         |        | ✓ | ✓ | ✓ | 26.1 |
|         | LiDAR  | - | - | ✓ | 44.2 |
|         |        | ✓ | - | ✓ | 45.7 |
|         |        | ✓ | ✓ | - | **46.6** |
|         |        | ✓ | ✓ | ✓ | 46.0 |

**Clustering Method.** To assess how clustering affects inlier–anomaly separation, we compare Support Vector Machine (SVM), K-Means, and an EM-based approach under identical quantization settings. As shown in Table 3, K-Means performs worse (43.5 mAP), suggesting that distance-based partitioning is sensitive to noisy activations and irregular object shapes. SVM and EM both reach 46.6 mAP; SVM benefits from a margin-based decision boundary that is less sensitive to local distortions, while EM introduces a probabilistic model over activation distributions, yielding a more principled decomposition. These results support our choice of EM as the default, as it combines competitive performance with a distribution-aware formulation.

Table 3: **Clustering methods.**

| Method | mAP |
|--------|-----|
| K-Means | 43.5 |
| SVM | **46.6** |
| EM (Ours) | **46.6** |

## 6 CONCLUSION

We presented Inlier-Centric Quantization (InlierQ), a post-training framework designed to tackle the inherent difficulty of quantizing object detection models under low-bit constraints. By explicitly separating inlier and anomaly activations, InlierQ suppresses spurious background signals while preserving object-relevant features, effectively compressing the dynamic range around meaningful activations. Experiments on COCO and nuScenes show consistent gains in both localization and classification accuracy, demonstrating that inlier-aware quantization enables robust and energy-efficient inference for 2D and 3D object detection models.

## REPRODUCIBILITY STATEMENT

To ensure reproducibility, we fully describe InlierQ, including the inlier–anomaly decomposition and the inlier-centric quantization objective, in Sections 3–4 and Appendix A. Implementation details (calibration, optimization, and quantization settings) are provided in Section 5. Experiments are conducted on public benchmarks (COCO and nuScenes) under standard evaluation protocols. Baselines are matched on model architectures and evaluation settings: BRECQ is run with its official implementation, whereas LiDAR-PTQ is reproduced by us following the paper and publicly available resources. Ablation studies (Section 5.4) isolate the contribution of each component.

## ACKNOWLEDGMENTS

This work was supported by Institute of Information & Communications Technology Planning & Evaluation(IITP) grant funded by the Korea government(MSIT) (RS-2024-00439020, Developing Sustainable, Real-Time Generative AI for Multimodal Interaction, SW Starlab).

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

## A  PROOF: EXPECTED HESSIAN OF THE LOSS EQ. (7) EQUALS THE FISHER INFORMATION MATRIX (FIM)

$$\mathcal{L}(\mathbf{x}) = -\frac{1}{KC} \sum_{k=1}^{K} \sum_{c=1}^{C} \log \mathcal{H}_{[k],c}(\mathbf{x}), \tag{13}$$

For simplicity, index each selected top-$K$ heatmap entry by a single index $i \in \{1, \dots, KC\}$ and write its value as $h_i(\mathbf{x}) \in [0, 1]$. We interpret $h_i(\mathbf{x})$ as a Bernoulli success probability.

**Pseudo-labeling.** Top-$K$ entries correspond to the model-predicted object locations. Assuming an object is always present at these predicted locations, we assign the pseudo-label $z_i = 1$ for all selected top-$K$ entries, and non-top-$K$ entries are treated as $z = 0$.

Thus, starting from the standard Negative Log-Likelihood (NLL) over all heatmap locations,

$$\ell(\mathbf{x}) = - \sum_j \Big[ z_j \log h_j(\mathbf{x}) + (1 - z_j) \log \big(1 - h_j(\mathbf{x})\big) \Big],$$

our pseudo-labeling assigns $z_j = 1$ on the selected top-$K$ entries and $z_j = 0$ on the remaining (non-top-$K$) entries. This yields

$$\ell(\mathbf{x}) = - \sum_{j \in \text{Top-}K} \log h_j(\mathbf{x}) \ - \sum_{j \notin \text{Top-}K} \log \big(1 - h_j(\mathbf{x})\big).$$

In eq. (13), we intentionally keep only the first term, i.e., we optimize *only* the top-$K$ predicted object locations and ignore the other non-top-$K$ heatmap queries. Equivalently, we restrict the likelihood to the top-$K$ subset and drop the contribution from $j \notin$ Top-$K$. Therefore, for each selected entry $i \in$ Top-$K$ we use $p(z_i = 1 \mid \mathbf{x}) = h_i(\mathbf{x})$, and the per-entry negative log-likelihood becomes

$$\ell_i(\mathbf{x}) = - \log p(z_i = 1 \mid \mathbf{x}) = - \log h_i(\mathbf{x}), \tag{14}$$

which matches the term used in eq. (13).

**Proof.** Fix one selected top-$K$ entry $i$ with Bernoulli model $p_i(z \mid \mathbf{x}) = h_i(\mathbf{x})^z \big(1 - h_i(\mathbf{x})\big)^{1-z}$, $z \in \{0, 1\}$. Since $\sum_z p_i(z \mid \mathbf{x}) = 1$, we have

$$\sum_z \nabla_{\mathbf{x}} p_i(z \mid \mathbf{x}) = \mathbf{0}.$$

Using $\nabla p = p \nabla \log p$ gives the zero-mean score:

$$\mathbb{E}_{z \sim p_i}\big[\nabla_{\mathbf{x}} \log p_i(z \mid \mathbf{x})\big] = \mathbf{0}.$$

Differentiating once more yields the information equality:

$$\mathbb{E}_{z \sim p_i}\Big[\big(\nabla_{\mathbf{x}} \log p_i(z \mid \mathbf{x})\big)\big(\nabla_{\mathbf{x}} \log p_i(z \mid \mathbf{x})\big)^{\top}\Big] = \mathbb{E}_{z \sim p_i}\Big[ - \nabla_{\mathbf{x}}^2 \log p_i(z \mid \mathbf{x})\Big].$$

Define the Fisher information matrix (FIM) for entry $i$ as

$$\mathbf{F}_i(\mathbf{x}) := \mathbb{E}_{z \sim p_i}\Big[\big(\nabla_{\mathbf{x}} \log p_i(z \mid \mathbf{x})\big)\big(\nabla_{\mathbf{x}} \log p_i(z \mid \mathbf{x})\big)^{\top}\Big].$$

Then $\mathbf{F}_i(\mathbf{x}) = \mathbb{E}_{z \sim p_i}\big[\nabla_{\mathbf{x}}^2 \ell_i(\mathbf{x}; z)\big]$ with $\ell_i(\mathbf{x}; z) := - \log p_i(z \mid \mathbf{x})$. Averaging over the selected $KC$ top-$K$ entries in eq. (13) shows that the expected Hessian equals the FIM.

