# OpenReview forum: "Inlier-Centric Post-Training Quantization for Object Detection Models"
_ICLR.cc/2026/Conference — ICLR 2026 Poster_

### Official Review · Reviewer_N4RF · 2025-10-29

**Soundness:** 3
**Presentation:** 3
**Contribution:** 2
**Rating:** 4
**Confidence:** 3

**Summary:**

The paper proposes InlierQ, a post-training quantization (PTQ) method tailored for object detection models. The approach introduces a saliency-driven inlier/anomaly decomposition to prioritize quantization precision for task-relevant activations while suppressing noisy or outlier features. InlierQ uses a gradient-based volume saliency score, fits an EM-based posterior over the score, and uses this to define inlier sets per layer. Empirical results on COCO and nuScenes benchmarks for 2D and 3D detection show that InlierQ offers modest but consistent reductions in quantization errors and improved accuracy over BRECQ and LiDAR-PTQ, especially at low bit-widths.

**Strengths:**

1. The central insight—distinguishing inlier from anomaly activations using a saliency score—addresses a notable gap in previous PTQ approaches that treat all activations equally. The probabilistic EM-based classification is straightforward yet effectively leverages gradient information.
2. Experiments span both 2D (COCO) and 3D (nuScenes) detection tasks with multiple modalities (camera and LiDAR), covering several state-of-the-art detection architectures. Results in Table 1 show consistent improvements of up to 2 mAP over BRECQ on challenging low-bit settings.
3. The derivation connecting the Hessian of the custom loss function to the Fisher Information Matrix provides a solid theoretical grounding.

**Weaknesses:**

1. A key ablation study is missing. I would like to see the performance of the quantization loss directly weighted by the saliency map.
2. In equation 7, the H is the k-th largest heatmap value. I cannot get the meaning of the $k$. The whole proof has no $k$ in it, then how is equation 7 equivalent to the FIM? The effect of $k$ should also be evaluated by ablation study.

**Questions:**

See weakness 2.

---

> ### Author Response · Authors · 2025-11-21
> **Response to Reviewer#4 N4RF**
>
> We sincerely thank the reviewer for the careful review of our work and for the insightful questions and concerns. Below, we respond to your comments in detail, addressing each point individually.
>
> > **W1.** “A key ablation study is missing. I would like to see the performance of the quantization loss directly weighted by the saliency map.”
>
> **Reply.** We appreciate the reviewer’s insightful recommendation. To address the concerns, we analyze three quantization variants: (i) activations weighted by the probability (saliency) map, (ii) unweighted and unfiltered activations (baseline), and (iii) anomaly-filtered activations (our inlier-only approach).
>
> | Method   | Detector    | Bits | mAP  |
> |:--------:|:-----------:|:----:|:----:|
> | Case. i  | CenterPoint | W4A4 | 46.0 |
> | Case. ii | CenterPoint | W4A4 | 44.2 |
> | Case. iii| CenterPoint | W4A4 | 46.6 |
>
>
> > **W2-1.** “In equation 7, the H is the k-th largest heatmap value. I cannot get the meaning of the $k$. The whole proof has no $k$ in it, then how is equation 7 equivalent to the FIM?”
>
> **Reply.** A heatmap query $\mathcal{H}_{[k],c}\in(0,1)$ is the k-th largest probability that a class-c object is present in a volume, which corresponds to the spatial location at coordinate $(u,v)$ for a 2D image or a BEV activation $\in\mathbb{R}^{H\times W\times C}$, and $(i,j,k)$ for a 3D voxel activation $\mathbb{R}^{D\times H\times W\times C}$.
> We model these heatmap probabilities with a Bernoulli likelihood and adopt the negative log-likelihood (NLL) so that the Hessian of the loss in Eq. (7) coincides with the FIM. To clarify this, we revised the corresponding derivation in the appendix of our manuscript.
>
>
> > **W2-2.** “The effect of  k should also be evaluated by ablation study.”
>
> **Reply.** As shown in the table below, a larger $K$ typically provides enriched information. Yet, a too-large $K$ can contain task-irrelevant heatmap samples, making a reasonable trade-off point at the training-time K configuration (highlighted in bold). We added this study as a plot in our revised manuscript Fig. 5. We thank the reviewer for their recommendation.
>
> | Detector | Bits | K=100 | K=200 | **K=300** | K=400 | K=500 |
> |:-----------:|:----:|:-----:|:-----:|:---------:|:-----:|:-----:|
> | DETR3D      | W4A4 | 24.8  | 25.2  | **26.4**  | 25.6  | 25.8  |
>
> | Detector | Bits | K=50 | K=250 | **K=500** | K=750 | K=1000 |
> |:-----------:|:----:|:----:|:-----:|:---------:|:-----:|:------:|
> | CenterPoint | W4A4 | 45.1 | 45.5  | **46.6**  | 46.0  | 45.9   |

---

> > ### Comment · Reviewer_N4RF · 2025-11-24
> >
> > I thank the authors for their detailed response and additional experiments. My concerns regarding the missing ablation study and the mathematical notation in Equation 7 have been satisfactorily resolved. specifically:
> >
> > 1.  The comparison between the saliency-weighted quantization and the proposed method (Response to W1) effectively demonstrates the advantage of the proposed approach.
> > 2.  The explanation of the definition of $k$ and its connection to the FIM, along with the sensitivity analysis of $K$ (Response to W2), clarifies the theoretical grounding and robustness of the method.
> >
> > Given that my major concerns have been addressed, I have decided to raise my score.

---

> ### Author Response · Authors · 2025-11-24
> **Official Comment by Authors**
>
> We sincerely appreciate your thoughtful review and the revised score. Your feedback has been invaluable in improving our work, and we are pleased that our response resolved your concerns and further reinforced the validity of our proposed method.

---

### Official Review · Reviewer_gEWM · 2025-10-30

**Soundness:** 3
**Presentation:** 2
**Contribution:** 3
**Rating:** 6
**Confidence:** 2

**Summary:**

InlierQ introduces an inlier centric PTQ framework that computes gradient aware volume saliency, classifies activations into inliers and outliers with EM, and concentrates quantization on the inlier subspace. It is label free, needs only 64 calibration samples, and consistently cuts quantization error while preserving detection accuracy on COCO and nuScenes for both camera and LiDAR detectors.

**Strengths:**

1. Conceptually novel with clear theoretical grounding, using an inlier-centric optimization that allocates bit precision to task-relevant activations.
2. Strong engineering practicality, since it is label-free and training-free, and as a plug-in PTQ module it needs only 64 calibration samples.
3. Broad applicability across modalities and architectures, covering camera-based 2D detection, camera-based 3D detection, and LiDAR-based 3D detection.

**Weaknesses:**

1. Gains are limited at higher bits. Under W8A8 the performance is close to full precision or baseline PTQ, so the advantage is less pronounced.
2. Sensitivity to hyperparameters and unresolved robustness questions. The threshold τ controls the inlier ratio and the final accuracy, which may require retuning across datasets and detection heads.
3. Modest improvement on 2D detection tasks. Ablations indicate that Inlier and Anomaly Sets are less separable in 2D, which reduces the benefit.

**Questions:**

Please address my concerns in Weaknesses.

---

> ### Author Response · Authors · 2025-11-21
> **Response to Reviewer#3 gEWM (1/2)**
>
> We are deeply grateful to the reviewer for the time and effort devoted to thoroughly evaluating our work and for the insightful questions and concerns raised. In the following, we respectfully provide detailed responses to your comments, addressing each point in turn.
>
> > **W1.** “Gains are limited at higher bits. Under W8A8 the performance is close to full precision or baseline PTQ, so the advantage is less pronounced.”
>
> **Reply.** We appreciate the reviewer’s comment. InlierQ is particularly effective at low-bit quantization, while yielding only modest gains at higher bit precision. This behavior is closely tied to the characteristics of anomalies.
>
> Specifically, as shown in Fig. 1., anomalies cause the expansion of quantization range, and skew the activation distribution towards task-irrelevant activations’ intensity. At higher 8-bit precision, both anomalies and inliers are sufficiently representable, so the additional benefit of removing anomalies is modest. However, the same range expansion and distributional skew severely degrades performance at lower 4-bit, where the few available levels cannot accurately cover the whole activation intensities.
>
> To further examine the effectiveness of InlierQ under extremely low-bit settings, we additionally investigate 3-bit activation quantization using W8A3 bit precisions:
>
> | 2D Detector  | Backbone     | Bits |   Method   |  mAP |
> |:------------:|:--------:|:----:|:----------:|:----:|
> | RetinaNet | RegNet-3.2GF | W8A3 | BRECQ      | 30.9 |
> | RetinaNet | RegNet-3.2GF | W8A3 | LiDAR-PTQ  | 31.7 |
> | RetinaNet | RegNet-3.2GF | W8A3 | Ours       | **33.1** |
>
> | 2D Detector  | Backbone     | Bits |   Method   |  mAP |
> |:------------:|:--------:|:----:|:----------:|:----:|
> | Faster R-CNN | ResNet50     | W8A3 | BRECQ      | 23.5 |
> | Faster R-CNN | ResNet50     | W8A3 | LiDAR-PTQ  | 10.2 |
> | Faster R-CNN | ResNet50     | W8A3 | Ours       | **31.9** |
>
> | 3D Detector  | Modality | Bits |   Method   |  mAP |
> |:------------:|:--------:|:----:|:----------:|:----:|
> | DETR3D       | Camera   | W8A3 | BRECQ      |  9.6 |
> | DETR3D       | Camera   | W8A3 | LiDAR-PTQ  | 18.9 |
> | DETR3D       | Camera   | W8A3 | Ours       | **23.8** |
>
> | 3D Detector  | Modality | Bits |   Method   |  mAP |
> |:------------:|:--------:|:----:|:----------:|:----:|
> | CenterPoint  | LiDAR    | W8A3 | BRECQ      | 43.6 |
> | CenterPoint  | LiDAR    | W8A3 | LiDAR-PTQ  | 41.9 |
> | CenterPoint  | LiDAR    | W8A3 | Ours       | **45.3** |
>
> As shown here, InlierQ consistently outperforms the baseline methods, demonstrating its effectiveness in the low-bit quantization.
>
> > **W2.** “Sensitivity to hyperparameters and unresolved robustness questions. The threshold τ controls the inlier ratio and the final accuracy, which may require retuning across datasets and detection heads.”
>
> **Reply.** We thank the reviewer for raising this concern. In practice, we observe that our method is not particularly sensitive to $\tau$. In all experiments, we fix a single threshold $\tau = 0.9$ across all datasets and detection heads, without per-model tuning. Even when we make the inlier definition stricter in the high-confidence regime ($0.9 < \tau < 0.99$), we do not observe severe performance degradation.
>
> We also note that Fig. 4, the $\tau$-sensitivity plot, was produced under standard Min-Max quantization. This setting does not exploit the tolerance of the optimization-based PTQ objective in Eq. (2) and is more vulnerable to task-irrelevant anomalies. To better reflect the robustness of InlierQ for $\tau$, we have re-plotted and changed Fig. 4 using the layer-wise optimization instead of simple Min-Max quantization.

---

> ### Author Response · Authors · 2025-11-21
> **Response to Reviewer#3 gEWM (2/2)**
>
> > **W3-1.** “Modest improvement on 2D detection tasks.”
>
> **Reply.** COCO benchmark images typically have a higher fraction of pixels occupied by objects than nuScenes, so the proportion of inliers can often exceed that of anomalies. As a result, the anomaly distribution is less dominant within the overall activation distribution, which mitigates the severity of anomaly-driven performance degradation in 2D detection.
>
> Nevertheless, our W4A4 experiments in Table. 1, show clear and consistent performance gains, which align with the lower-bit effectiveness of InlierQ. Our investigated table in the response of W1 further support its applicability, showing larger task performance attains compared to the baseline methods.
>
>
> > **W3-2.** “Ablations indicate that Inlier and Anomaly Sets are less separable in 2D, which reduces the benefit.”
>
> **Reply.** As shown in Fig. 1, even when we restrict activations to the anomaly set, their distributions remain quite similar to that of the inlier set. Therefore, the performance gaps in Table 2 alone didn't fully reveal how well inliers and anomalies are separated.
> Instead, we re-plot Fig. 4 to directly examine the inlier and anomaly sets obtained by probability thresholding. In Fig. 4, as we increase the inlier-probability threshold, the task performance consistently improves, whereas increasing the anomaly-probability threshold consistently degrades performance. This opposite behavior provides clear evidence that the inlier and anomaly sets are meaningfully separable in terms of task relevance.

---

> > ### Comment · Reviewer_gEWM · 2025-11-27
> >
> > The authors have resolved my main concerns, so I will keep my original score.

---

> > > ### Author Response · Authors · 2025-11-27
> > > **Official Comment by Authors**
> > >
> > > We greatly appreciate the reviewer’s acknowledgement of our clarifications and are pleased that our response has addressed the concerns raised in the original review. The reviewer's careful consideration has been invaluable in enhancing the clarity and rigor of our presentation of the proposed methods.

---

### Official Review · Reviewer_pWqE · 2025-11-01

**Soundness:** 2
**Presentation:** 3
**Contribution:** 2
**Rating:** 4
**Confidence:** 3

**Summary:**

The paper proposes InlierQ, a post-training quantization approach that separates activations into inliers and anomalies using a “volume saliency score.” The authors claim that existing quantization methods treat all activations uniformly, thus failing to account for this distinction.  The idea of this paper is to allocate quantization bit capacity to informative activations (inliers) while suppressing noisy or anomalous ones. Experiments on COCO (2D detection) and nuScenes (3D detection) show consistent but moderate performance improvements.

**Strengths:**

- The method is clearly described with equations and an algorithmic flow (Algorithm 1), making the paper easy to follow and reproduce.
- The results cover both 2D and 3D object detection models (COCO, nuScenes), showing robustness under different modalities and architectures.
- The authors identify that quantization error can be dominated by high-magnitude anomalies or uninformative background activations, which is indeed an important real-world problem for low-bit quantization in detection models. In practice, InlierQ can be applied as a drop-in PTQ refinement module without retraining.

**Weaknesses:**

- The claim that existing quantization approaches treat all activations uniformly is inaccurate. A substantial body of prior work has explicitly or implicitly modelled activation importance. Although the authors mention outlier-suppression methods such as SmoothQuant, QDrop, and SVDQuant in Related Work, the distinction they claim is that these works only relax amplitudes while their method decomposes activations into inliers and anomalies.

However, many existing PTQ methods already model activation importance and distribution heterogeneity through gradient-weighted or low-rank mechanisms (e.g., BRECQ, Adaround, SVDQuant). Thus, I think the proposed “inlier decomposition” represents a rephrase of known ideas rather than a novel quantization paradigm.

- The EM-based decomposition into “inliers” and “anomalies” in the Method Section is heuristic and lacks theoretical justification. Thus, it does not introduce a new optimization or probabilistic framework beyond existing adaptive scaling methods.

- The improvements (mAP on COCO / nuScenes) are modest and within the range of normal variance. Thus, it is unclear whether the improvement arises from the “inlier” modelling or simply from additional regularization during calibration. No ablation on this factor is performed.

- While some baselines (SmoothQuant, QDrop) are mentioned in related work, they are not included in experiments. The claim of superiority over “outlier suppression” is not directly supported.

- It is unclear how InlierQ performs under extremely low bitwidth (e.g., 2–3 bits) when compared to adaptive rounding or Hessian-based PTQ.

**Questions:**

- Is the inlier detection performed per-layer, per-channel, or per-sample?

---

> ### Author Response · Authors · 2025-11-21
> **Response to Reviewer#2 pWqE (1/3)**
>
> We sincerely thank the reviewer for the valuable feedback. In particular, the suggestion to discuss extremely low-bit quantization helped us better highlight our contributions. Below, we respond to the reviewer’s comments in detail.
>
> > **W1-1.** “The claim that existing quantization approaches treat all activations uniformly is inaccurate. A substantial body of prior work has explicitly or implicitly modelled activation importance.”
>
> **Reply.** We thank the reviewer for pointing out the wording that may potentially mislead the readers. We intended to highlight the significant practical impact of disruptive anomalies, rather than implying that all methods rely on uniform quantization.
> We revised the sentence on line 42 as:
>
> “Although such disruptive anomalies should be filtered out, the lack of a principled way to identify them hinders addressing the problem. This causes the quantization range to be easily dominated by task-irrelevant activations, thereby undermining the preservation of task-relevant features.”
>
> In addition to this, we updated Fig. 1 and Fig. 3 to improve the presentation of our problem statement and methods.
>
>
> > **W1-2.** “Although the authors mention outlier-suppression methods such as SmoothQuant, QDrop, and SVDQuant in Related Work, the distinction they claim is that these works only relax amplitudes while their method decomposes activations into inliers and anomalies. However, many existing PTQ methods already model activation importance and distribution heterogeneity through gradient-weighted or low-rank mechanisms (e.g., BRECQ, Adaround, SVDQuant). Thus, I think the proposed “inlier decomposition” represents a rephrase of known ideas rather than a novel quantization paradigm.”
>
> **Reply.** We appreciate for the reviewer’s insightful comments. We believe that the sentence can be misleading. The distinction of inlierQ is to introduce an inlier posterior as a probabilistic measure of task relevance and use it to define a decision boundary between task-relevant inliers and disruptive anomalies.
>
> This approach is particularly necessary for object detection, where sensing noise, background clutter, and unrelated foreground signals produce redundant activations dominating the activation distributions. As shown in Fig. 1, anomalies inflate the intensity range and skew a distribution towards task-irrelevant activations, due to their large number of samples and the wide range of intensities. We address this issue by explicitly separating the anomalies from the layer-wise calibration for quantization.
>
> To correct the presentation, we revised the sentences in line 107,
>
> from:
> “Rather than relaxing anomalies in their arrangement and concentration, outlier suppression targets amplitude. Even after peaks are attenuated, range shift can persist due to anomaly effects, which motivates us to decompose activations into inliers and anomalies via a saliency score.”
>
> to:
> “As shown in Fig. 1, anomalies redundantly inflate the quantization range and induce distributional skew, which in turn hinders accurate quantization. This motivates us to decompose activations into inliers and anomalies via a volume saliency score, and reject the samples for an intrinsic addressing.”
>
> **Novelty.** InlierQ explicitly isolates anomaly distributions from activations to mitigate their adverse impact on quantization. While prior methods aim to reduce quantization error over all activations, our approach concentrates the optimization objective on inliers. This inlier-centric mechanism, in turn, preserves information in task-relevant activations, especially under low activation bit precisions.
>
> With this unique property, InlierQ can serve as a plug-and-play auxiliary module for quantizing object detection models and can be combined with existing approaches, simultaneously (1) compressing quantization ranges, (2) correcting skewed activation distributions, yet (3) preserving salient information.

---

> ### Author Response · Authors · 2025-11-21
> **Response to Reviewer#2 pWqE (2/3)**
>
> > **W2.** “The EM-based decomposition into “inliers” and “anomalies” in the Method Section is heuristic and lacks theoretical justification. Thus, it does not introduce a new optimization or probabilistic framework beyond existing adaptive scaling methods.”
>
> **Reply.** Our inlier–anomaly criterion is theory-driven rather than heuristic, and inlierQ introduces learning of layer-wise decision boundaries for anomaly separation. An inlier is defined as an activation whose inlier probability exceeds a high confidence level ($\tau$ in Eq. 11), and any activation that does not satisfy this condition is treated as an anomaly. From a Bayesian decision-theoretic perspective, the inlier probability summarizes the evidence that an activation belongs to the task-relevant component, and thresholding it at $\tau$ corresponds to a classification decision boundary between inliers and anomalies.
>
> As shown in Fig. 1, restricting quantization to the inlier set yields a more compressed quantization range and a reshaped intensity distribution compared to using the full activation samples, yet it typically improves detection performance. This observation indicates that the proposed inlier-centric criterion effectively identifies the task-relevant distribution and is a key factor behind the observed gains.
>
> > **W3.** “The improvements (mAP on COCO / nuScenes) are modest and within the range of normal variance. Thus, it is unclear whether the improvement arises from the “inlier” modelling or simply from additional regularization during calibration. No ablation on this factor is performed.”
>
> **Reply.** Inlier-centric quantization is inherently more effective at lower activation bit precisions without any regularizations. Under 8-bit activation quantization, existing methods already have sufficient capacity to represent both inliers and anomalies, so the additional benefit of removing anomalies is limited. In contrast, with 4-bit activations, the few available levels can be disproportionately spent on anomalies, forcing task-relevant inliers into very coarse bins. By filtering out anomalies and reallocating these levels to inliers, InlierQ naturally yields much larger gains in the 4-bit regime than in the 8-bit regime, which supports the inlier-centric design.
>
> We also note that we used a standard min–max scheme for weight quantization and InlierQ is specifically designed for activation quantization; therefore, the modest gain observed in the W4A8 setting does not reflect a limitation of InlierQ.
> To solidify the effectiveness of inlierQ for low-bit, we investigated 3-bit activation quantization in the answer of W5, and revised the related explanations in line 362,
>
> from:
> “Overall, these results show that all methods preserve accuracy under mild quantization (e.g., W8A8), whereas our approach remains robust even when quantization is restricted to the compressed, de-skewed inlier activations in aggressive low-bit settings (e.g., W4A4). Preserved or even improved accuracy under this restriction indicates that the rejected anomalies carry little task-relevant information and supports the proposed inlier set as a safe and effective target for activation quantization, highlighting the benefit of explicitly modelling activation distributions.
> ”
>
> to:
> “
> Overall, these results show that quantizing activations on the compressed, de-skewed inlier distributions preserves and often improves accuracy compared to the baselines, especially under 4-bit activations. This indicates that the rejected anomalies carry little task-relevant information and that the proposed inlier set is a safe target for quantization. In the 8-bit regime, the gains are modest, whereas in the more aggressive 4-bit setting, they become pronounced because representing the entire activation distribution, including anomalies, is much harder and consumes a disproportionate fraction of the limited quantization levels. This behavior directly supports the inlier-centric design, where removing anomalies is particularly beneficial at low bit precision.
> ”

---

> ### Author Response · Authors · 2025-11-21
> **Response to Reviewer#2 pWqE (3/3)**
>
> > **W4.** “While some baselines (SmoothQuant, QDrop) are mentioned in related work, they are not included in experiments. The claim of superiority over “outlier suppression” is not directly supported.”
>
> **Reply.** In the related work section, we review outlier-suppression methods to motivate explicitly addressing anomalies and to clarify how anomalies differ from conventional outliers.
>
> > **W5.** “It is unclear how InlierQ performs under extremely low bitwidth (e.g., 2–3 bits) when compared to adaptive rounding or Hessian-based PTQ.”
>
> **Reply.** We thank you for your recommendation. We investigated the 3-bit quantization for object detection models. For the 3D detectors DETR3D and CenterPoint, performance was severely corrupted under 3-bit activation quantization, so we quantized only the backbone networks (ResNet-101 for DETR3D and VoxelNet for CenterPoint). As shown in the table below, our main claim still holds in this low-bit regime, with our method consistently outperforming the baseline approaches.
>
> | 2D Detector  | Backbone     | Bits |   Method   |  mAP |
> |:------------:|:--------:|:----:|:----------:|:----:|
> | RetinaNet | RegNet-3.2GF | W8A3 | BRECQ      | 30.9 |
> | RetinaNet | RegNet-3.2GF | W8A3 | LiDAR-PTQ  | 31.7 |
> | RetinaNet | RegNet-3.2GF | W8A3 | Ours       | **33.1** |
>
> | 2D Detector  | Backbone     | Bits |   Method   |  mAP |
> |:------------:|:--------:|:----:|:----------:|:----:|
> | Faster R-CNN | ResNet50     | W8A3 | BRECQ      | 23.5 |
> | Faster R-CNN | ResNet50     | W8A3 | LiDAR-PTQ  | 10.2 |
> | Faster R-CNN | ResNet50     | W8A3 | Ours       | **31.9** |
>
> | 3D Detector  | Modality | Bits |   Method   |  mAP |
> |:------------:|:--------:|:----:|:----------:|:----:|
> | DETR3D       | Camera   | W8A3 | BRECQ      |  9.6 |
> | DETR3D       | Camera   | W8A3 | LiDAR-PTQ  | 18.9 |
> | DETR3D       | Camera   | W8A3 | Ours       | **23.8** |
>
> | 3D Detector  | Modality | Bits |   Method   |  mAP |
> |:------------:|:--------:|:----:|:----------:|:----:|
> | CenterPoint  | LiDAR    | W8A3 | BRECQ      | 43.6 |
> | CenterPoint  | LiDAR    | W8A3 | LiDAR-PTQ  | 41.9 |
> | CenterPoint  | LiDAR    | W8A3 | Ours       | **45.3** |
>
> > **Q1.** “Is the inlier detection performed per-layer, per-channel, or per-sample?”
>
> **Reply.** It is performed per-layer and per-sample (pixel-wise for 2D and voxel-wise for 3D). For each layer $l$, a layer-specific classifier $f^l: \mathbb{R}^1\mapsto$ {$inlier, anomaly$} is learned separately, and maps the volume saliency score $h(\mathbf{x})\in \mathbb{R}^1$ of a C-dimensional activation vector $\mathbf{x}\in \mathbb{R}^C$ to an inlier/anomaly label as: $f^l (h(\mathbf{x})) \in$ {$inlier, anomaly$}.

---

> ### Author Response · Authors · 2025-11-28
> **Official Comment by Authors**
>
> Again, we thank the reviewer for the time and effort dedicated to our work. During the rebuttal period, we sought to fully address your concerns and further substantiate the validity of our method, yet due to unforeseen circumstances, it is unfortunate that we are not able to fully engage in this discussion.
>
> Regardless of the final decision, we would sincerely appreciate any further comments you may share. We are truly grateful for the time and consideration you have invested in our submission.

---

### Official Review · Reviewer_t1mq · 2025-11-02

**Soundness:** 3
**Presentation:** 2
**Contribution:** 2
**Rating:** 6
**Confidence:** 3

**Summary:**

1. Motivation: Quantization is a crucial technique for model compression. Existing quantization methods treat all activations uniformly, but in object detection tasks, background activations (task-irrelevant activations) are both numerous and can exhibit anomalously high values. This uniform treatment leads to substantial waste of bit precision, especially under low-bit quantization settings.

2. Method: The authors propose Inlier-Centric Quantization (InlierQ). First, they compute volume saliency scores based on gradients and then use a Gaussian Mixture Model (GMM)-based posterior probability to partition the feature space into inlier and outlier regions. Quantization optimization is applied exclusively to the inlier region. To further emphasize task-relevant features, the authors design a top-K heatmap-based loss, which concentrates the Hessian computation on the most discriminative channels.

3. Experiments: Experimental results demonstrate that InlierQ achieves superior performance in both 2D and 3D object detection quantization tasks, particularly under low-bit settings.

**Strengths:**

1. The paper is presented fairly clearly.

2. The motivation is well-articulated, and the proposed InlierQ method is reasonably designed.

**Weaknesses:**

1. In Equation (7), is the supervision applied to the top-$K$ entries for each channel of the heatmap? If so, the summation indices over $K$ and $C$ might be reversed in the equation.

2. Equation (12) is described as “explicitly discards anomalous activations and focuses only on the curvature of inlier distributions.” Does this imply that in Equation (6), $\lambda_I = 1$ and $\lambda_O = 0$? If so, by directly discarding background activations and focusing only on high-gradient regions, could the quantization range be overly compressed? Might this lead to an increase in false positive detections in the quantized model? The authors should ideally provide experimental results on precision to support this claim.

3. In the experimental setup, the authors do not specify the backbones used. Referring to experiments in BRECQ and AQD, it is recommended that the method be evaluated on additional detection algorithms (e.g., RetinaNet) and more backbones (e.g., MobileNetV2) to demonstrate its generality.

**Questions:**

As shown in Weakness.

---

> ### Author Response · Authors · 2025-11-21
> **Response to Reviewer#1 t1mq (1/2)**
>
> We thank the reviewer for the time and effort dedicated to carefully reviewing our work and for raising insightful questions and concerns. In the following, we provide detailed responses to your comments, addressing each point one by one.
>
> > **W1.** “In Equation (7), is the supervision applied to the top- entries for each channel of the heatmap? If so, the summation indices over  and  might be reversed in the equation.”
>
> **Reply.** Thank you for the careful feedback. We apply the channel-wise top-K selection before the summation in Equation (7). As the resulting index sets are finite and fixed, the k- and c-summations can be interchanged without changing the value.
>
>
> > **W2.** “Equation (12) is described as “explicitly discards anomalous activations and focuses only on the curvature of inlier distributions.” Does this imply that in Equation (6),  and ? If so, by directly discarding background activations and focusing only on high-gradient regions, could the quantization range be overly compressed? Might this lead to an increase in false positive detections in the quantized model? The authors should ideally provide experimental results on precision to support this claim.”
>
> **Reply.**  We first fit a two-component (inlier and anomaly) Gaussian mixture to the activation statistics using the EM algorithm. After convergence, we interpret the inlier posterior probability as an inlier probability model and define the inlier set by thresholding this posterior as in Eq. (11). In our updated visualizations in Fig. 1, the resulting inlier set exhibits a compressed quantization range while still reducing the overall quantization error, which in turn leads to improved task performance.
>
> In response to the reviewer’s concerns, we compare three metrics for Faster R-CNN with a ResNet-50 backbone: the total number of false positives over the COCO dataset at a 50% IoU threshold (#FP), average recall (AR; higher values indicate fewer false negatives), and average precision (AP; higher values indicate fewer false positives) under 8-, 4-, and 3-bit activation quantization with 8-bit Min-Max weight quantization, against the baseline methods. We also note that BRECQ optimizes over all activations, so its performance directly reflects the negative impact of anomalies.:
>
> | Bits | Method |  # FP |   AR |   AP |
> |------|--------|------:|-----:|-----:|
> | W8A8 | BRECQ  | 136K | 52.4 | **37.8** |
> | W8A8 | Ours   | **135K** | **52.5** | **37.8** |
>
> | Bits | Method |  # FP |   AR |   AP |
> |------|--------|------:|-----:|-----:|
> | W8A4 | BRECQ  | 169K | 50.6 | 35.3 |
> | W8A4 | Ours   | **142K** | **51.4** | **36.8** |
>
> | Bits | Method |  # FP |   AR |   AP |
> |------|--------|------:|-----:|-----:|
> | W8A3 | BRECQ  | 408K | 38.9 | 23.5 |
> | W8A3 | Ours   | **192K** | **47.8** | **32.1** |

---

> > ### Comment · Reviewer_t1mq · 2025-11-26
> >
> > The authors have addressed my concerns. I will maintain my original score.

---

> ### Author Response · Authors · 2025-11-21
> **Response to Reviewer#1 t1mq (2/2)**
>
> >**W3.** “In the experimental setup, the authors do not specify the backbones used. Referring to experiments in BRECQ and AQD, it is recommended that the method be evaluated on additional detection algorithms (e.g., RetinaNet) and more backbones (e.g., MobileNetV2) to demonstrate its generality.”
>
> **Reply.** We appreciate the reviewer’s recommendation. Our Faster R-CNN experiments use ResNet-50 as the backbone. In response to the reviewer’s request, we have extended Table 1-(a) (2D detection) by adding RetinaNet with a RegNet-3.2GF backbone, and we further report additional results below, for a range of backbones combined with either RetinaNet or Faster R-CNN heads:
>
> | Detector     | Bits | Method     | ResNet-18 | ResNet-50 | RegNet-800MF | RegNet-3.2GF |
> |:------------:|:----:|:----------:|:---------:|:---------:|:------------:|:------------:|
> | RetinaNet    | FP32 | -          | 31.7      | 36.5      | 35.6         | 39.0         |
> | RetinaNet    | W8A8 | BRECQ      | **31.7**  | 36.3      | **35.6**     | 38.9         |
> | RetinaNet    | W8A8 | LiDAR-PTQ  | 31.6      | **36.4**  | 35.4         | **39.0**     |
> | RetinaNet    | W8A8 | Ours       | **31.7**  | **36.4**  | **35.6**     | **39.0**     |
>
> | Detector     | Bits | Method     | ResNet-18 | ResNet-50 | RegNet-800MF | RegNet-3.2GF |
> |:------------:|:----:|:----------:|:---------:|:---------:|:------------:|:------------:|
> | RetinaNet    | FP32 | -          | 31.7      | 36.5      | 35.6         | 39.0         |
> | RetinaNet    | W4A8 | BRECQ      | 27.9      | 34.4      | **33.2**     | **36.4**     |
> | RetinaNet    | W4A8 | LiDAR-PTQ  | 27.9      | 34.4      | **33.2**     | 36.3         |
> | RetinaNet    | W4A8 | Ours       | **28.0**  | **34.5**  | **33.2**     | **36.4**     |
>
> | Detector     | Bits | Method     | ResNet-18 | ResNet-50 | RegNet-800MF | RegNet-3.2GF |
> |:------------:|:----:|:----------:|:---------:|:---------:|:------------:|:------------:|
> | RetinaNet    | FP32 | -          | 31.7      | 36.5      | 35.6         | 39.0         |
> | RetinaNet    | W4A4 | BRECQ      | 22.8      | 31.6      | 30.9         | 34.0         |
> | RetinaNet    | W4A4 | LiDAR-PTQ  | 22.7      | 33.1      | 31.0         | 34.4         |
> | RetinaNet    | W4A4 | Ours       | **25.0**  | **33.4**  | **31.2**     | **34.7**     |
>
> | Detector     | Bits | Method     | ResNet-18 | ResNet-50 | RegNet-800MF | RegNet-3.2GF |
> |:------------:|:----:|:----------:|:---------:|:---------:|:------------:|:------------:|
> | Faster R-CNN | FP32 | -          | 33.9      | 37.9      | 35.1         | 39.9         |
> | Faster R-CNN | W8A8 | BRECQ      | **33.9**  | **37.8**  | **35.1**     | **39.8**     |
> | Faster R-CNN | W8A8 | LiDAR-PTQ  | 33.8      | 37.7      | **35.1**     | 39.7         |
> | Faster R-CNN | W8A8 | Ours       | **33.9**  | **37.8**  | **35.1**     | **39.8**     |
>
> | Detector     | Bits | Method     | ResNet-18 | ResNet-50 | RegNet-800MF | RegNet-3.2GF |
> |:------------:|:----:|:----------:|:---------:|:---------:|:------------:|:------------:|
> | Faster R-CNN | FP32 | -          | 33.9      | 37.9      | 35.1         | 39.9         |
> | Faster R-CNN | W4A8 | BRECQ      | 30.1      | 36.0      | 32.9         | **37.9**     |
> | Faster R-CNN | W4A8 | LiDAR-PTQ  | 30.1      | 36.0      | 32.8         | 37.8         |
> | Faster R-CNN | W4A8 | Ours       | **30.2**  | **36.1**  | **33.1**     | **37.9**     |
>
> | Detector     | Bits | Method     | ResNet-18 | ResNet-50 | RegNet-800MF | RegNet-3.2GF |
> |:------------:|:----:|:----------:|:---------:|:---------:|:------------:|:------------:|
> | Faster R-CNN | FP32 | -          | 33.9      | 37.9      | 35.1         | 39.9         |
> | Faster R-CNN | W4A4 | BRECQ      | 26.0      | 32.7      | 29.8         | 32.5         |
> | Faster R-CNN | W4A4 | LiDAR-PTQ  | 26.5      | 34.3      | 31.4         | 35.5         |
> | Faster R-CNN | W4A4 | Ours       | **26.9**  | **34.7**  | **31.6**     | **36.1**     |

---

> ### Author Response · Authors · 2025-11-27
> **Official Comment by Authors**
>
> We sincerely thank the reviewer for acknowledging our clarifications. We are pleased that our response has addressed the concerns raised in the original review.

---

### Comment · Area_Chair_W6WV · 2025-11-26
**A Reminder on Your Crucial Role in the ICLR Discussion Period**

Dear Reviewers who haven't engaged with the rebuttal:

As the Area Chair, I would like to sincerely thank you for the time and expertise you have invested in writing your initial review. Your insights are invaluable to the decision-making process.

We are now entering the critical discussion and rebuttal phase. This is a collaborative process where authors have the opportunity to address your concerns and questions. Your active participation in this phase is essential to ensure we reach a fair and well-informed final decision.

I strongly encourage you to:

Engage with the Authors' Rebuttal: Please read the authors' response carefully and substantively.

Participate in the Discussion: Engage with the other reviewers on the forum. If the authors have clarified a point, please acknowledge it. If you have follow-up questions or remaining concerns, please voice them. Your dialogue with fellow reviewers is key to reaching a consensus.

Update Your Review (if necessary): Based on the discussion and rebuttal, you may feel the need to adjust your score or final recommendation. Please do so, as it reflects a more holistic view of the paper.

Your continued engagement ensures the integrity and quality of the ICLR conference. Thank you for your vital contribution to our community.

Best regards,

Area Chair, ICLR 2026

---

### Author Response · Authors · 2025-12-03
**Author Final Remarks**

Dear Area Chair and Reviewers,

We deeply thank the reviewers for their time and insightful feedback. Below, we briefly summarize the core contributions of our work and how our revisions addressed the reviewers’ main concerns.

---

## 1. Key Contributions

• Identify and address quantization error induced by task-irrelevant anomalies. Reviewer pWqE (Strengths 3) highlighted this as an important real-world issue in low-bit quantization, and Reviewer N4RF (Strengths 1) noted that our formulation leverages gradient information in a clear and effective way.

• Introduce a probabilistic separation of task-relevant inliers from anomalies and concentrate quantization on the inlier subspace. Reviewer gEWM (Strengths 1) described this inlier-centric optimization as conceptually novel and theoretically grounded, while Reviewer N4RF (Strengths 1) regarded it as an effective saliency-driven inlier/anomaly decomposition.

• Offer a principled yet plug-and-play PTQ regime. Reviewer t1mq (Strengths 1–2) found the method well-motivated and reasonably designed. Reviewer pWqE (Strengths 1) commented that it is easy to follow, reproduce, and use as a drop-in refinement, and Reviewer gEWM (Strengths 2) emphasized its practical appeal as a label-free, drop-in approach requiring only a few calibration samples.

---

## 2. Discussion Summary

• Reviewer t1mq questioned the formulation of Eq. (7) and Eq. (12), the safety of discarding anomalies, and the generality across detectors and backbones. We clarified the equations and inlier definition, added False Positive/Averaged Recall/Averaged Precision analysis, and extended 2D object detection experiments to RetinaNet with ResNet-18/50 and RegNet-800MF/3.2GF, which showed reduced false positives and stable gains; the reviewer kept a positive score (6).

• Reviewer pWqE raised concerns about novelty relative to importance-aware PTQ and requested evidence in more aggressive low-bit regimes. We revised the text to modestly position our contribution as a probabilistic inlier posterior with layer-wise decision boundaries, and added 3-bit activation results on 2D/3D object detection models (RetinaNet, Faster R-CNN, DETR3D, CenterPoint), where InlierQ consistently shows notable performance gains compared to our baseline work, BRECQ.

• Reviewer gEWM asked about limited gains at higher bits and 2D detection task, and robustness to $\tau$. We clarified the larger gains observed in low-bit and 3D detection task settings, and provided a revised $\tau$-sensitivity analysis for a clearer presentation of inlierQ. This revision addressed the reviewer's main concerns, and the score remained positive (6).

• Reviewer N4RF requested ablations and a clearer connection between the heatmap formulation and the FIM. We added ablations comparing saliency-weighted, unweighted, and inlier-only objectives, and refined the FIM-based derivation with a K-sensitivity study. These revisions addressed the reviewer’s concerns, and the score was raised to 6.

---

## 3. Conclusion

In summary, the discussion phase helped us clarify the probabilistic inlier-centric design, strengthen the empirical evaluation in low-bit settings, and provide additional ablations on anomaly rejection. We hope these revisions make the method and its scope clearer for modern low-bit object detection.

---

Best regards,

The Authors of #11042.

---

### Meta-Review · Area_Chair_8uZg · 2026-01-06

**Summary:**

This paper proposes Inlier-Centric Quantization (InlierQ), a post-training quantization framework tailored for object detection models. The key idea is to explicitly distinguish task-relevant inlier activations from disruptive anomalies using a gradient-aware volume saliency score and an EM-based probabilistic decomposition, and to focus quantization optimization on the inlier subspace. Extensive experiments on both 2D and 3D object detection benchmarks (COCO and nuScenes), across camera- and LiDAR-based detectors, show that InlierQ consistently reduces quantization error and improves detection accuracy, particularly under low-bit activation quantization (4-bit and 3-bit), where existing PTQ methods struggle.

**Reviewer Concerns:**

The main concerns raised by reviewers centered on (i) the clarity and novelty of the inlier/anomaly decomposition relative to prior PTQ methods, (ii) the theoretical justification of the EM-based inlier modeling, (iii) the modest gains at higher bit-widths, and (iv) missing experimental details and ablations, including precision metrics, backbone diversity, extreme low-bit settings, and sensitivity analyses. The authors addressed these concerns comprehensively in the rebuttal.

**Reviewer Scores:**

The submission received multiple marginal-to-positive scores, with two reviewers rating it 6 (marginally above acceptance threshold).  Importantly, several reviewers explicitly acknowledged that their major concerns were resolved after the rebuttal and additional experiments, with one reviewer is willing to raise his/her score accordingly.

---

### Decision · Program_Chairs · 2026-01-26

Accept (Poster)